

# Isolating the climate change impacts on air pollution-related-pathologies over Europe - A modelling approach on cases and costs

Patricia Tarín-Carrasco[1], María Morales-Suárez-Varela[2,3], Ulas Im[4], Jørgen Brandt[4], Laura Palacios-Peña[1], and Pedro Jiménez-Guerrero[1,5]

[1]Department of Physics, Regional Campus of International Excellence Campus Mare Nostrum, University of Murcia, Murcia, Spain
[2]2Unit of Public Health and Environmental Care, Department of Preventive Medicine, University of Valencia, Valencia, Spain
[3]CIBER Epidemiology and Public Health (CIBERESP), Madrid, Spain
[4]Aarhus University, Department of Environmental Science, Frederiksborgvej 399, DK-4000, Roskilde, Denmark
[5]Biomedical Research Institute of Murcia (IMIB-Arrixaca), 30120 Murcia, Spain

**Correspondence:** Pedro Jiménez-Guerrero (pedro.jimenezguerrero@um.es)

**Abstract.**

Air pollution has important implications on human health and associated external costs to society, and is closely related to climate change. This contribution tries to assess the impacts of present (1996-2015) and future (2071-2100 under RCP8.5) air pollution on several cardiovascular and respiratory pathologies and to estimate the difference in the costs associated to those

health impacts on European population. For that, air quality data from the WRF-Chem regional chemistry/climate modelling system is used, together with some epidemiological information from the European Commission. The methodology considered relies on the EVA exposure-response functions and economic valuations (Brandt et al., 2013a, b). Several hypothesis have been established, in order to strictly isolate the effects of climate change on air pollution and health: constant present-day emission levels and population density in all Europe. In general, the number of cases for the pathologies considered will increase in the

future (chronic bronchitis, heart failure, lung cancer, premature deaths), increasing the overall cost associated from 173 billion €/year to over 204 billion €/year at the end of the present century. Premature deaths are the most important problem in the target area in terms of costs (158 billion €per year, increasing by 17% in the future RCP8.5 2071-2100 projection) and cases (418 700 cases/year, increasing by 94 900 cases/year in the future). The most affected areas are European megacities, the Ruhr Valley and several cities at eastern Europe (e.g. Chisinau, Bucharest). For the RCP8.5 scenario, cases and costs will increase

over southern and eastern Europe, while central and northern Europe could benefit by climate change variations (decreasing both cases and costs for the studied pathologies).

## 1 Introduction

Nowadays, air pollution is a serious environmental concern with a severe impact on population: on the one hand, by its close relationship with climate change; and on the other hand, because of its effects on human health and welfare. Air pollution is



an environmental problem affecting the entire planet, either by local or transboundary pollution (Ravishankara et al., 2012). In 2012, 3.7 million of premature deaths were caused by exposition to air pollution worldwide (WHO, 2013). In addition, indirectly, air pollution, has external costs to society related to damages to human health. For this reason, the control of emissions of atmospheric pollutants and having reliable future air quality estimations can represent a good strategy for mitigating

air pollution-related pathologies (Lelieveld et al., 2015). However, European targets for emissions and air pollution are not reached in southern Europe not only because of anthropic emissions, but also by natural causes (Pozzer et al., 2012), being the Mediterranean Basin the most affected area by increases in air pollution in present and future climate scenarios (Colette et al., 2012; Jiménez-Guerrero et al., 2013a).

Quantifying premature death caused by air pollution is difficult for several reasons. First, the lack of monitoring stations;

second, the variable toxicity of pollutants depending on their nature (Lelieveld et al., 2015). The effects of pollutants on the population also depend on its composition, exposition time and health condition of dwellers. Life habits must be as well considered: for instance, Stieb et al. (2017) recommended to reduce exposure time and physical activity outdoor under episodes with high concentrations of air pollutants. Moreover, the diverse combination of air pollutants can have additives or synergists adverse effects on health (Curtis et al., 2006). Another factor hampering the attribution of deaths or different pathologies caused

by air pollution is its association with high temperatures (Pearce et al., 2016). These authors indicate the relationship between high temperature short-term exposition and mortality risk. Therefore, heat extreme events are another aspect to contemplate for population health under a changing climate.

Pollutants of largest concern for human health in Europe are particulate matter (PM) with a diameter lower than 2.5 microns (PM2.5), nitrogen oxides ($NO_x$), sulfur dioxide ($SO_2$), tropospheric ozone ($O_3$) and carbon monoxide (CO) (Pozzer et al.,

2012). Both $O_3$ and PM are related with cardiorespiratory diseases and premature death. The most important pollutant for the mortality is PM, with an important decrease on the life expectancy projected for future scenarios (Héroux et al., 2015). PM exposure, especially to fine particles (PM2.5), may severely affect human health (Brook et al., 2010), piercing up lungs or even pulmonary alveoli (Pope and Dockery, 2006). They produce cardiorespiratory symptoms and illness, increased asthma cases, heart attacks, stroke, and even premature mortality (Tagaris et al., 2010). Fine particles can produce damage even in

small concentrations (Beelen et al., 2014). Giannadaki et al. (2017) estimated premature deaths caused by PM2.5 at 3.15 million/year in 2010 globally, while their estimation for Europe was around 173 000 premature deaths (about 5% of the global rate). Gaseous pollutants like $NO_x$ or $SO_2$ may get in the organism by inhalation and affect respiratory system, irritating the respiratory system, and inducing bronchoconstriction and asthma (Kampa and Castanas, 2007).

Recently, Im et al. (2018) estimated the health impacts of air pollution in Europe and the United States (US) by using

concentration inputs from different chemistry-transport models in the Economic Valuation of Air Pollution (EVA) system (Brandt et al., 2013a, b). In Europe, the total number of premature deaths (acute and chronic) and associated costs is calculated to be 414 000 and 300 billion€, respectively.

In addition, climate change alone may affect the concentration of these gaseous pollutants and particles through modifications on chemistry on gaseous phase, transport, deposition and natural emissions (Jacob and Winner, 2009). Modelling





approaches (together with remote sensing) may represent a good methodology to disentangle the role of climate change on air pollution and get future projections of air quality (Jerrett et al., 2017).

Due to changes on the future climatic conditions, air quality will importantly worsen, especially in southern Europe (Jiménez-Guerrero et al., 2013a). Several studies have combined atmospheric science, epidemiology, public health and economy and tries
to asses future air pollution, mitigation strategies and its relation and repercussions on population health and associated costs. For instance, Geels et al. (2015) indicate that climate change effect together with a reduction of emissions will decrease the premature deaths caused by air pollution. These authors estimated a reduction for acute mortality caused by PM in a 36%-64% for 2050s and 53%-84% for 2080s; and a decrease of 62%-65% and almost 80% for same future periods if chronic mortality is targeted. Héroux et al. (2015) suggest that mortality risk associated to air pollution can be reversible on a short period, as a
year.

Henceforth, this study focuses on the analysis of population health problems caused by regulatory pollutants in Europe, and their associated costs. First, the methodology followed tries to get the correlation (if any) between particles with a diameter under 10 microns (PM10) and total deaths and deaths caused by respiratory diseases over Europe, by using epidemiological data from the European Commission for the period 2001-2012. Then, in order to assess the impact of air pollution on health
for present and future scenarios, data from a regional chemistry/climate model for a present climatology (1996-2015) will be used for estimating the cases and associated costs of some related pathologies and the differences found for a future climate scenario (2071-2100, RCP8.5).

## 2  Methodology

### 2.1  Epidemiological study for present-climate situation

First, an epidemiological study for present situation has been carried out, with data obtained from the European Commission (Eurostat) (https://ec.europa.eu/eurostat/statistics-explained/index.php?title=Health) corresponding to the years 2001-2012. Total Death (TD) and Death caused by Respiratory Diseases (DRD) have been analysed. The objective is to search the relationship between such mortalities and air pollution (in our study case, PM10, due to the short time series available for PM2.5). Although mortality data was available since 1994, the targeted period begins in 2001 due to the availability of PM10
data.

This study covers 25 European countries in total, with a non-homogeneous time coverage because the different year of entry into the European Union. Taking into account the data availability, the countries selected for the epidemiology analysis were Austria, Belgium, Bulgaria, Czech Republic, Denmark, Estonia, Finland, France, Germany, Hungary, Iceland, Ireland, Italy, Luxembourg, The Netherlands, Norway, Poland, Portugal, Romania, Slovakia, Slovenia, Spain, Sweden, Switzerland and
United Kingdom.

The correlation is not done directly on the raw data, but the mortality and PM10 series are detrended in order to avoid spurious correlations. With the anomaly series, a correlation between total mortality and PM10 and between deaths caused by



respiratory disease and such pollutant is carried out. The correlations found have undergone a Mann-Kendall test in order to assure their significance at 95% confidence (p<0.05).

## 2.2 Regional chemistry/climate simulations

In addition, air quality model data is used in order to check the possible changes in pathologies and diseases between present
and future scenarios of climate change. The simulations used for assessing air quality in this work span the periods 1996-2015, as a present reference period, and 2071-2100 under the RCP8.5 scenario, as a future enhanced forcing scenario. This scenario is at the top of radiative forcing scenarios among all the Representative Concentration Paths (Moss et al., 2010). The differences between these two runs (present and RCP8.5) will provide the changes in future air quality.

The regional chemistry/climate model used has been WRF-Chem (Grell et al., 2005). The spatial model configuration
comprises a domain covering most of southern and central Europe with a resolution of 25 km for WRF-Chem simulations. Thirty-three sigma levels are considered in the vertical, with the top of the atmosphere at 50 hPa. Historical simulations with WRF-Chem (1996-2015) were driven by the ERA20C reanalysis (Poli et al., 2016), whose approximate resolution is 125 km; and the 200-km resolution CMIP5-experiment r1i1p1 MPI-ESM-LR (Taylor et al., 2012; Giorgetta et al., 2012a). No nudging was conducted on the experiments. The CMIP5-experiment RCP8.5-forced r1i1p1 MPI-ESM-LR run (Giorgetta et al., 2012b)
was used for the scenario period (2071-2100)

Further information on the physico-chemical configuration of the model can be found in the scientific literature (Forkel et al., 2015; Palacios-Peña et al., 2017). A short description is presented here. The WRF-Chem setup used in these simulations include the following options: RADM2 chemical mechanism; MADE/SORGAM aerosol module including some aqueous reactions; Fast-J photolysis scheme; RRTMG shortwave and longwave radiation schemes; Yonsei University PBL scheme (YSU) for
the Planetary Boundary Layer; dry deposition follows the Wesely resistance approach, while wet deposition is divided into convective wet deposition and grid-scale wet deposition.

The modelling system for present-day climatologies has been extensively evaluated (Brunner et al., 2015). Despite the model skills with respect to air pollution modelling data used for health estimations are widely discussed(Im et al., 2018), more information with respect to ozone and particulate matter (PM10) can be found in Im et al. (2015a, b).

In order to isolate the possible effects of climate change alone on air pollutants, unchanged anthropogenic emissions coming from ACCMIP (Lamarque et al., 2010) are assumed. That allows to anticipate the possible impacts if no mitigation strategies for regulatory pollutants are carried out and characterize the climate penalty on air quality levels. Natural emissions depend on climate conditions, and therefore vary in present and future simulations. Hence, the effects of climate change on air pollution follow the methodology explained in (Jiménez-Guerrero et al., 2013b), excluding possible changes in vegetation or land use.

## 2.3 Present and future impacts of air quality on pathologies

The impact of air quality on the following pathologies is contemplated in this work: Respiratory Hospital Admissions (RHA), Cerebrovascular Hospital Admissions (CHA), Congestive Heart Failure (CHF), Chronic Bronchitis (CB), Lung Cancer (LC)



**Table 1.** Pathology, exposure-response coefficients and economic valuation, taken from Brandt et al. (2013a).

| Pathology | Exposure-response coefficient | Valuation |
|---|---|---|
| Respiratory Hospital Admissions (RHA) | $3.46 \times 10^{-6}$ cases / $\mu$g m$^{-3}$ PM | |
| | $+ 2.04 \times 10^{-6}$ cases / $\mu$g m$^{-3}$ SO$_2$ | 7931€/case |
| Cerebrovascular Hospital Admissions (CHA) | $8.42 \times 10^{-6}$ cases / $\mu$g m$^{-3}$ PM | 10047€/case |
| Congestive Heart Failure (CHF) | $3.09 \times 10^{-5}$ cases / $\mu$g m$^{-3}$ PM | |
| | $+ 5.64 \times 10^{-7}$ cases / $\mu$g m$^{-3}$ CO | 16409€/case |
| Chronic Bronchitis (CB) | $8.20 \times 10^{-5}$ cases / $\mu$g m$^{-3}$ PM | 52962€/case |
| Lung Cancer (LC) | $1.26 \times 10^{-5}$ cases / $\mu$g m$^{-3}$ PM | 21152€/case |
| Premature Deaths (PD) | $3.27 \times 10^{-6}$ SOMO35 cases / $\mu$g m$^{-3}$ | |
| | $+ 7.85 \times 10^{-6}$ cases / $\mu$g m$^{-3}$ SO$_2$ | 2111888€/case |
| | $+ 1.138 \times 10^{-3}$ YOLL/ $\mu$g m$^{-3}$ (>30 yr) | 77199€/YOLL |

and Premature Deaths (PD), this latter related both to acute mortality and chronic mortality as defined in (Brandt et al., 2013a, b).

For that, the gridded population data has been obtained from the SocioEconomic Data and Applications Center (SEDAC) of NASA (http://sedac.ciesin.columbia.edu) at a resolution of 1 km$^2$ and interpolated to the working grid. Since the time coverage of our analysis is 1996-2015, the Population Density, v4 dataset for the year 2005 has been used, based on counts consistent with national censuses and population registers. The population by cell is shown in Fig. 1. For the future scenario, the population has been kept constant in order to have an educated guess of the possible impacts due only to changes in air quality.

In order to calculate the air quality impacts on the aforementioned pathologies, the methodology described in the EVA system (Brandt et al., 2013a, b), and references therein, has been used. In this work, we have utilized the population from SEDAC together with the WRF-Chem simulations and the exposure-response functions and economic valuations (as 2006 euros) compiled in Brandt et al. (2013a) to estimate external costs of air pollution. The exposure-response coefficient and the valuation used are compiled in Table 1.

## 3 Results and Discussion

### 3.1 Statistical epidemiological study for present situation

In the following section the results from collected data from Eurostat have been analysed. After detrending the data and calculating the anomalies for Total Death (TD), Deaths by Respiratory Diseases (DRD) and PM10, correlations for each country between mortality and particles have been established. The obtained results are shown on Table 2, where bold values indicate correlations significant at 95% confidence interval (p<0.05). Countries as Germany, Hungary, Italy and Slovenia





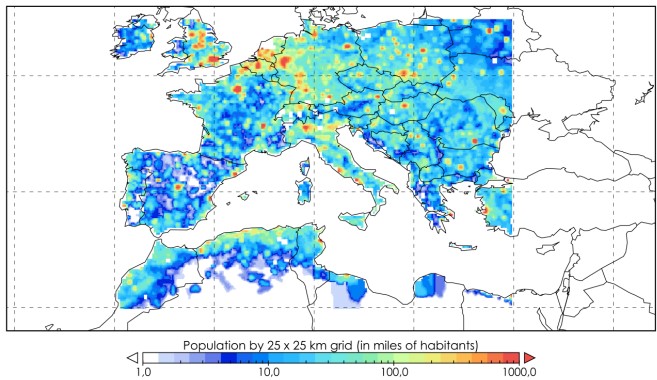

**Figure 1.** Population in each grid cell (in miles of habitants) (SEDAC population dataset for 2005).

present a clear relation between such pollutant and both TD and DRD, with a high correlation in the anomaly series and high statistical significance. Meanwhile, countries as Czech Republic, Estonia or Switzerland present a notable statistically-significant correlation only for TD-PM10 correlation. For the rest of the countries, mainly due to the short time series of data, either no significance (e.g. Bulgaria, Denmark, France, United Kingdom, Spain, Iceland) or a significant very low correlation

(e.g. Austria, The Netherlands, Sweden) is obtained.

Hence, for several countries (despite the short data series) we can establish a relationship between mortality (especially, TD) and atmospheric PM10 levels. Generally, the correlation TD-PM10 is higher than for DRD-TD. As pointed out by several authors, the justification for these higher correlation values can be found in mortality by PM10 being caused other pathologies that are not just respiratory, like cardiac, cerebrovascular, etc. (Curtis et al., 2006; Tagaris et al., 2010; Pozzer et al., 2012).

**3.2    Present and future scenarios study for the pathologies and costs related to air pollution**

This section discusses the results found for cases distribution (number of people) with different pathologies caused by several air pollutants for a present climate (1996-2015) and the differences with future RCP8.5 scenario (2071-2100). A summary of the global cases and associated costs is shown in Table 3.

**3.2.1    Respiratory Hospital Admissions (RHA)**

The results for the European domain targeted (Table 3) indicate 16400 cases of RHA per year in the 1996-2015 period, which will increase by 3800 cases in RCP8.5 2071-2100 scenario. The external costs associated to this pathology represent 87.1 M€ in the present climate, increasing in the future scenario by 20.5 M€ (that is, an increase in cases and costs of +23%).

If local differences in RHA are searched (Fig. 2), the highest number of present cases is found in the north of the study area, in countries such Belgium, Netherlands or in the western zones of Germany. Several hotspots appear in Paris and Bucharest,



**Table 2.** Correlations data between Total Death (TD) and PM10 (left column) and Deaths by Respiratory Diseases (DRD) and PM10 (right column) for European countries. Bold values indicate a significant correlation (p<0.05).

| Country | TD-PM10 | DRD-PM10 |
|---|---|---|
| Austria | **0.047** | **0.313** |
| Belgium | **0.071** | 0.051 |
| Bulgaria | -0.363 | -0.225 |
| Czech Republic | **0.455** | **0.313** |
| Denmark | **0.114** | -0.261 |
| Estonia | **0.407** | -0.391 |
| Finland | **0.168** | -0.155 |
| France | 0.230 | 0.272 |
| Germany | **0.522** | **0.512** |
| Hungary | **0.492** | **0.678** |
| Iceland | -0.387 | -0.649 |
| Ireland | -0.104 | **0.240** |
| Italy | **0.508** | **0.747** |
| Luxembourg | 0.503 | -0.105 |
| Netherlands | **0.086** | **0.128** |
| Norway | -0.020 | 0.400 |
| Poland | -0.125 | 0.095 |
| Portugal | -0.579 | -0.571 |
| Romania | -0.146 | -0.340 |
| Slovakia | **0.258** | **0.273** |
| Slovenia | **0.525** | **0.418** |
| Spain | 0.322 | 0.243 |
| Sweden | **0.199** | 0.195 |
| Switzerland | **0.351** | **0.169** |
| United Kingdom | -0.034 | -0.050 |

with an average of 200 cases per year and cell in 1996-2015. The cases with the lowest values are found in northern Spain and central France, with less than 0.25 cases/year cell. With respect to the future (2071-2100) differences, and despite the global increase of the cases for Europe, strong differences appear between southern and northern Europe. Over central/northern Europe a slight decrease in RHA cases is projected (up to -4 cases/year cell) on localized cities located in Germany -Berlin- or

5  Austria -Vienna-, meanwhile RHA cases may increase up to 122 per year and cell in southern and eastern Europe.

The costs follow the same spatial pattern as the cases, as expected. Despite the economic impacts on the society are limited, for several European megacities as Paris, Cologne or Bucharest the present cost of RHA may reach up to 1 M€. For the future



**Table 3.** Mean number of cases (in miles of cases) as associated costs (in million €) per year for each pathology for present climate conditions (1996-2015) and variations in the future RCP8.5 scenario (2071-2100) for the entire domain of simulation.

| Pathology | Cases ($\times 10^3$) (1996-2015) | $\Delta$Cases ($\times 10^3$) (2071-2100) | Costs (M€) (1996-2015) | $\Delta$Costs (M€) (2071-2100) |
|---|---|---|---|---|
| Respiratory Hospital Admissions (RHA) | 16.4 | +3.8 | 87.1 | +20.5 |
| Cerebrovascular Hospital Admissions (CHA) | 31.9 | +7.9 | 214.8 | +53.1 |
| Congestive Heart Failure (CHF) | 117.1 | +28.9 | 1 288.3 | +318.3 |
| Chronic Bronchitis (CB) | 310.6 | +76.8 | 11 982 | +2 962.8 |
| Lung Cancer (LC) | 47.7 | +11.8 | 764.7 | +189.1 |
| Premature Deaths (PD) | 418.7 | +94.9 | 158 970 | +27 346.0 |

scenario, external costs are expected to increase in various European areas, especially on eastern Europe, area which barely had costs associated in the present climatology; or in southern countries like Spain and Italy. In this latter areas, the costs increase in more than 0.5 M€ per year and cell for 2071-2100 with respect to the 1996-2015 period.

### 3.2.2 Cerebrovascular Hospital Admission (CHA)

Table 3 indicate a total number of 31900 cases of RHA per year in the 1996-2015 period (external cost of 214.8 M€), with an associated increase for all the domain of 7900 cases in RCP8.5 2071-2100 scenario (increase in costs: 53.1 M€). That is, overall cases and costs will increase by +25% at the end of the XXI century with respect to the present situation.

Regarding spatial distributed CHA, up to 400 cases per year are found in the city of Bucharest (Fig. 3), with an associated external cost of 2.5 M€. Many of the European megacities exceed the 100 cases for the present period. The countries with highest admission numbers are Belgium, The Netherlands and Germany. On the other hand, the northern half of Iberian Penin-

sula is the area least affected by CHA pathology. With respect to the differences with 2071-2100 RCP8.5 scenario, the spatial pattern follows the same structure as for RHA cases shown in Fig. 2, as previously commented: a general increase in southern Europe (up to 270 cases per year and cell) and a light decrease mainly in cities of The Netherlands, Germany and Austria (up to 10 less cases per year and cell). The increase in costs in the future scenario can reach up to +2 M€ in large cities such as Madrid, Bucarest or Paris, while the decrease of the costs in areas with reduced CHA such as Berlin or Vienna do not exceed

the -0.5 M€.

### 3.2.3 Congestive Heart Failure (CHF)

CHF, as pointed out in Table 3, involves 117100 cases per year over Europe in the 1996-2015 period, with an associated external cost of 1.3 billion €. Future climate change will increase the cases of CHF by +24% (28900 cases for the entire domain), also

increasing the associated costs by the same percentage (increase of 318.3 M€in external costs for the period 2071-2100).



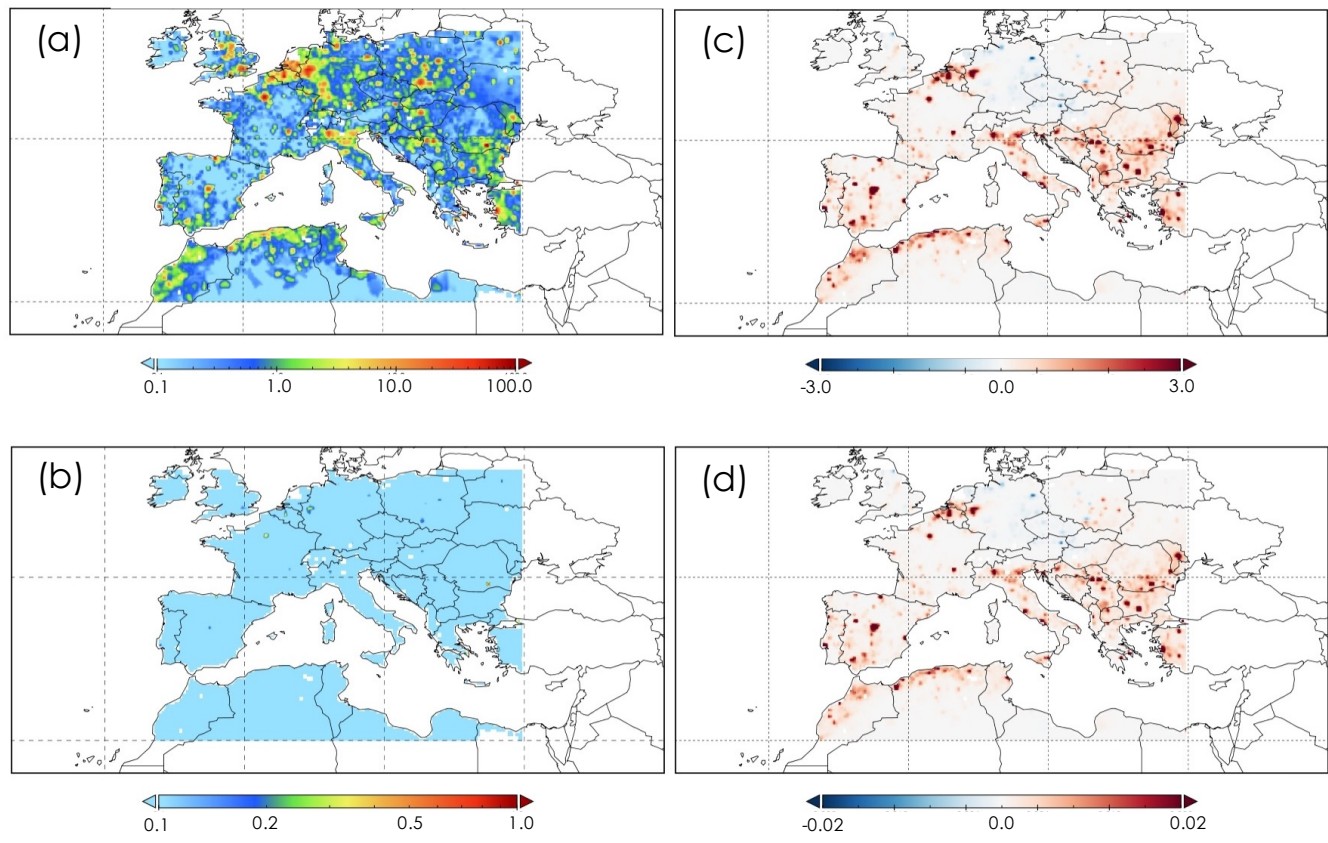

**Figure 2.** (a) Present cases by Respiratory Hospital Admissions (RHA) and (b) associated costs, in M€. (c) Changes projected in RHA cases and (d) changes in costs (M€) under the RCP8.5 scenario (2071-2100).

With respect to the spatial distribution of CHF (Fig. 4) within the target area, once again most of the CHF cases are located in Belgium and the Ruhr area; however punctual hotspots appear in the largest European cities (London, Paris, Madrid) with over 1000 cases per year in the entire cities (costs >10 M€), being the highest number found over the city of Bucharest (over 1500 cases per year in all the city, with an associated cost of over 16 M€). CHF cases are widely distributed throughout the territory, with values generally under 10 cases per year and cell. The area with the lowest number of CHF cases for 1996-2015 is the northern Iberian Peninsula. For future projections, an increase of CHF close to 1000 cases in Bulgaria (Sofia and Craiova) and Moldova (Chisinau), with an associated increase in costs up to 11 M€. Once again, a decrease in cases (-36 cases/year cell) and associated costs (variations of -0.4 M€/year cell) is found in central Europe cities such as Vienna or Berlin for the 2071-2100 RCP8.5 scenario.




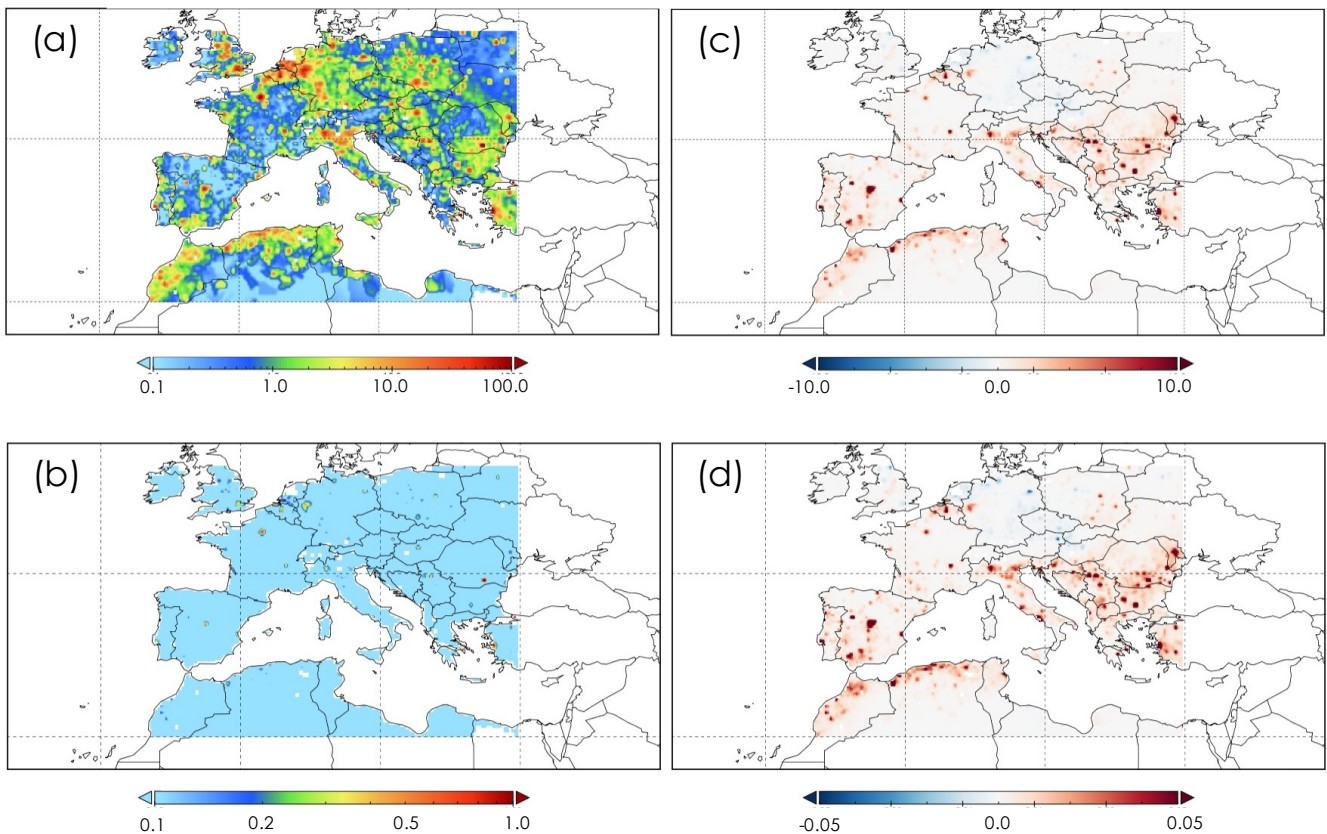

**Figure 3.** (a) Present cases by Cerebrovascular Hospital Admissions (CHA) and (b) associated costs, in M€. (c) Changes projected in CHA cases and (d) changes in costs (M€) under the RCP8.5 scenario (2071-2100).

### 3.2.4 Chronic Bronchitis (CB)

Table 3 indicates that CB cases exceed 310600 cases per year in all the target domain for the 1996-2015 period (cost 12 billion €), which will increase by 76800 cases in the RCP8.5 scenario (2071-2100), also increasing the cost by 3.0 billion €in all Europe covered by the simulation domain (+25%).

5    The CB cases are unevenly distributed over Europe (Fig. 5). This pathology is distributed throughout the study area analogously to CHA (Fig. 3), since the exposure-response coefficient depends for CB and CHA only on the concentration of particulate matter. Cases exceed 1000 per year (over 50 M€) in cities as Madrid, Paris, Brussels or Bucharest, with a maximum for the latter of 3892 cases for present period study (costs of 150 M€per year). Areas as northern Italy or eastern Europe are largely affected in comparison with other pathologies previously mentioned. For the future scenario, the most affected areas





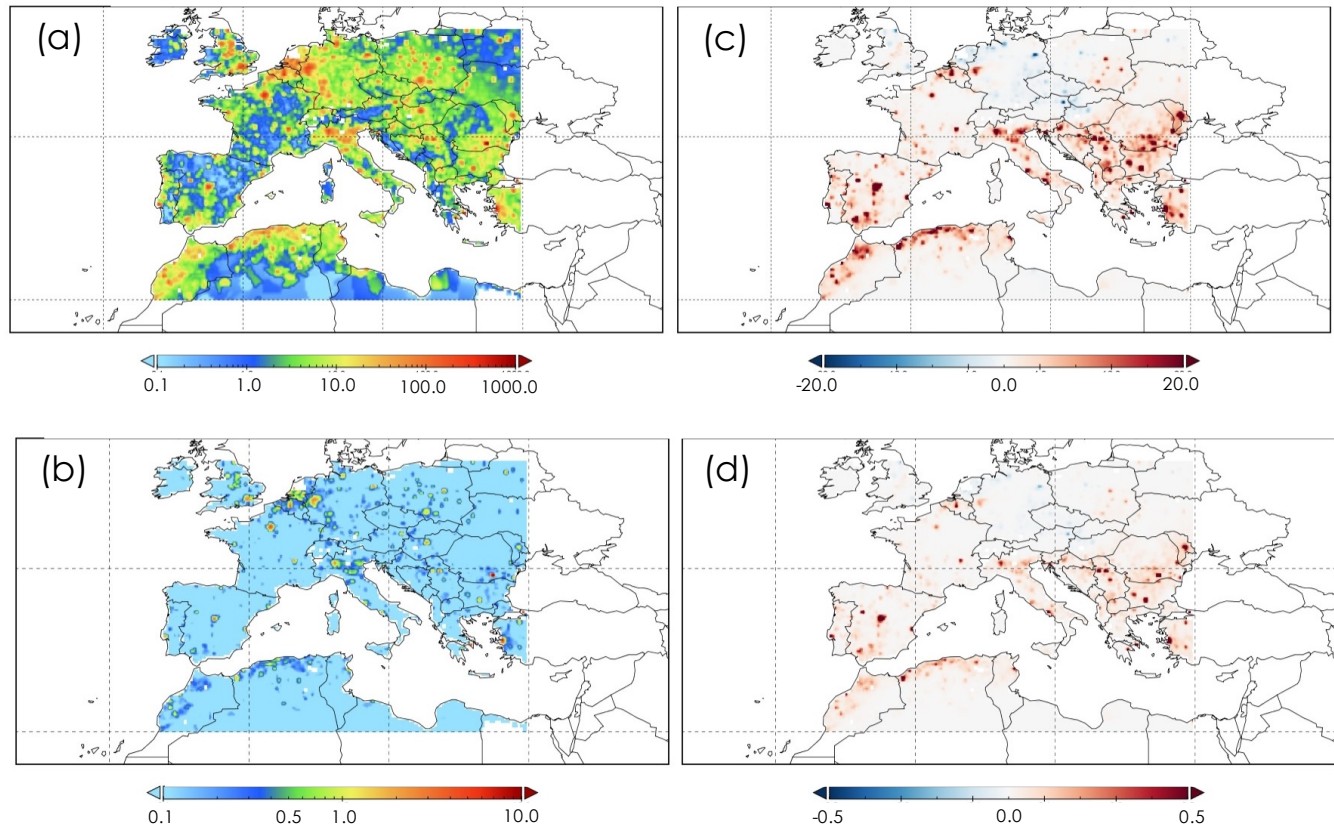

**Figure 4.** (a) Present cases by Congestive Heart Failure (CHF) and (b) associated costs, in M€. (c) Changes projected in CHF cases and (d) changes in costs (M€) under the RCP8.5 scenario (2071-2100).

are those reported for the present, with the exception of Chisinau (Moldova), Bucarest (Romania) and Sofia (Bulgaria), where future increases in CB cases may exceed 2600 (increase in external costs of over 100 M€), meanwhile some cities of Germany, Austria or Czech Republic could decrease these pathology cases up to almost 100 (-4 M€/year cell decrease in costs).

### 3.2.5 Lung Cancer (LC)

5   Regarding LC, Table 3 estimates 47700 cases per year in Europe for the present climatology, with an associated cost of 765 M€). The projected increase by 2071-2100 reaches +11800 extra cases in the RCP8.5 scenario, with an increase in costs of +189 M€(+25% of cases and cost increase).

    The results for the spatial distribution of LC (Fig. 6) indicate that LC affects principally central and northern Europe, with widespread hotspots throughout the territory. The maxima are found over European megacities (600 cases/year cell in the



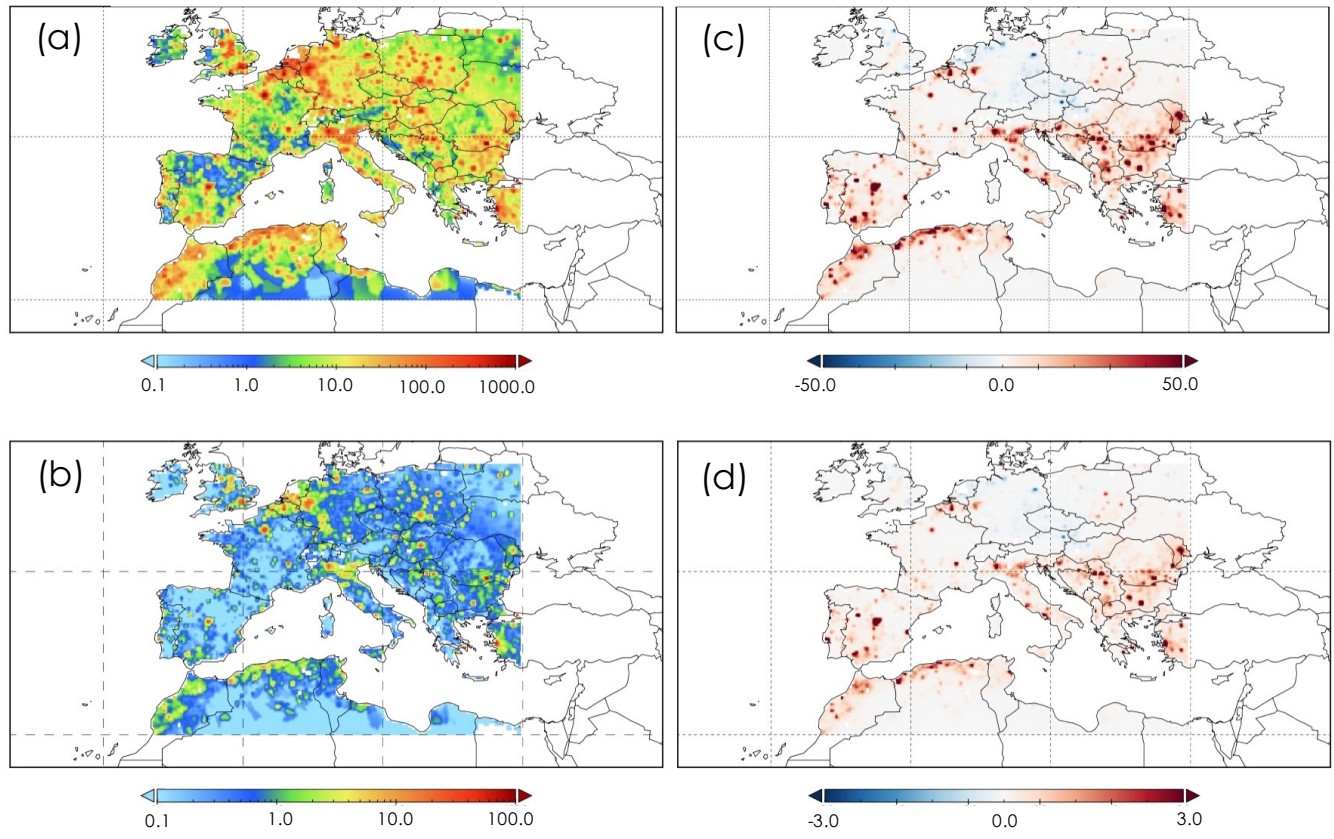

**Figure 5.** (a) Present cases by Chronic Bronquitis (CB) and (b) associated costs, in M€. (c) Changes projected in CB cases and (d) changes in costs (M€) under the RCP8.5 scenario (2071-2100).

present climatology; costs over 10 M€). Countries with a widest area with affectations are Belgium and The Netherlands, with an important number of cases in Germany, northern Italy or southern Poland. An increase of over 400 LC cases per year and cell (associated increase in cost of 6.5 M€/year cell) is expected for the 2071-2100 period over southern Europe (Madrid, Rome, Bucarest, Sofia or Belgrade), with decreases of around 10 cases/year cell in cities of eastern Germany and eastern Austria.

5  **3.2.6   Premature Deaths (PD)**

Last, the variable PD covers chronic mortality and acute mortality as defined in (Brandt et al., 2013a). Chronic mortality refers to mortality risks associated with long-term exposure, and is quantified in years of life lost (YOLL, depending on PM2.5 concentration for population >30 yr). Last, acute mortality depends on $SO_2$ levels and SOMO35 (Sum of Ozone Means Over 35 ppb), which is estimated as the sum of means over 35 ppb for the daily maximum 8-hour values of ozone.





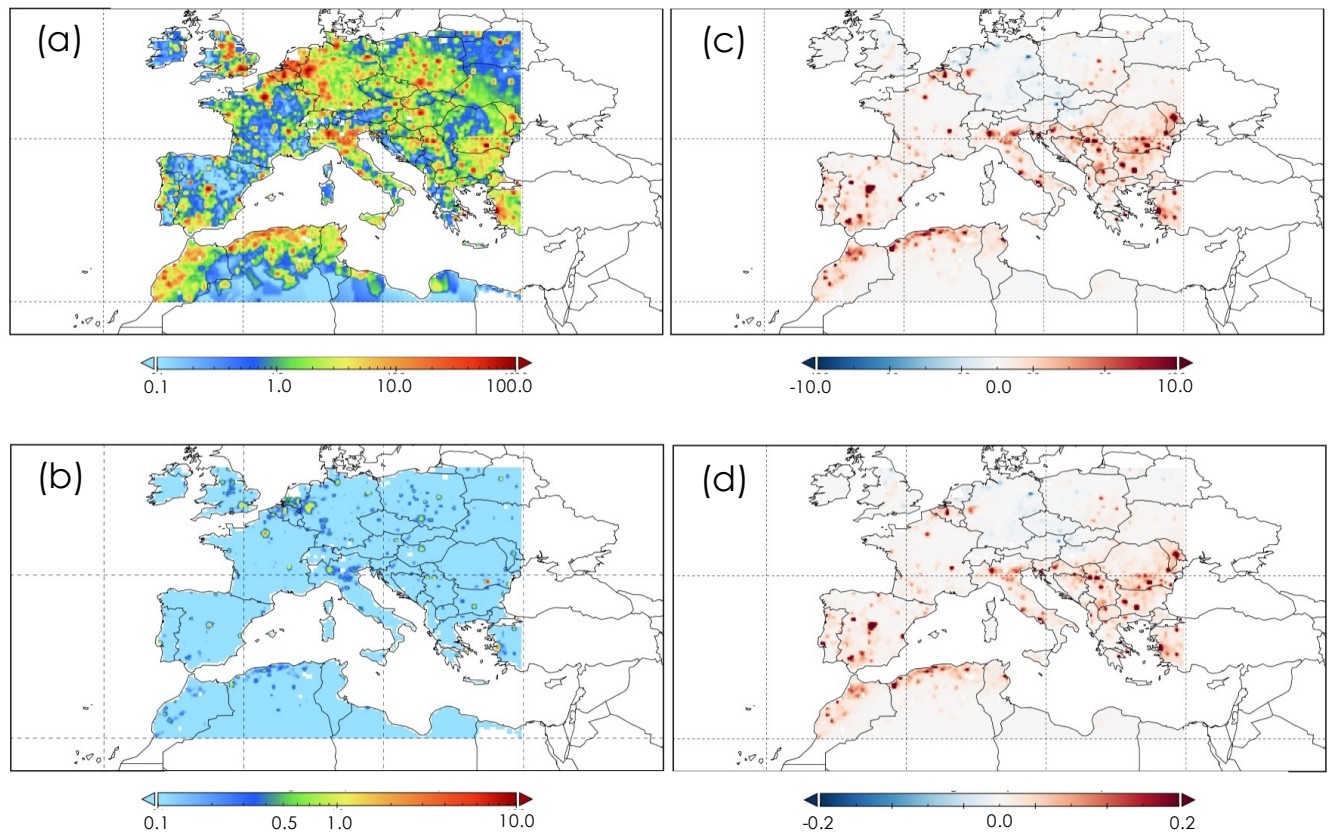

**Figure 6.** (a) Present cases by Lung Cancer (LC) and (b) associated costs, in M€. (c) Changes projected in RHA cases and (d) changes in costs (M€) under the RCP8.5 scenario (2071-2100).

Estimates of 418700 cases per year in the target domain are provided in Table 3 for 1996-2015, with a huge associated cost (159 billions €). The projected increase in the RCP8.5 for the years 2071-2100 reaches +94900 extra cases; that is, an increase in costs of over 27 billions € (+17% of cases and cost increase).

The dominant pathology over the entire domain (Fig. 7) is PD, especially over central Europe, Belgium, The Netherlands, Germany, Poland, Italy or Bulgaria. These countries have a high number of cases for the whole country. Hotspots are again located over large cities, exceeding 1000 cases/year cell and even reaching 4314 cases in several cities like Paris and London (associated external cost over 700 M€ in these cities). For the future scenario (2071-2100) a clear difference between the northern half and the southern half of study area is depicted. While in southern cities as Madrid (Spain) or eastern Europe cities as Belgrade (Serbia), Bucarest (Romania) or Sofia (Bulgaria) PD may increase up to 2400 cases per year (+450 M€ in several





megacities). In cities of countries as Germany (Berlin, Hamburg), France (Paris) or United Kingdom (London, Manchester, Newcastle) a decrease of more than 200 cases per year and cell are projected (reduction of costs over 31.5 M€/year cell).

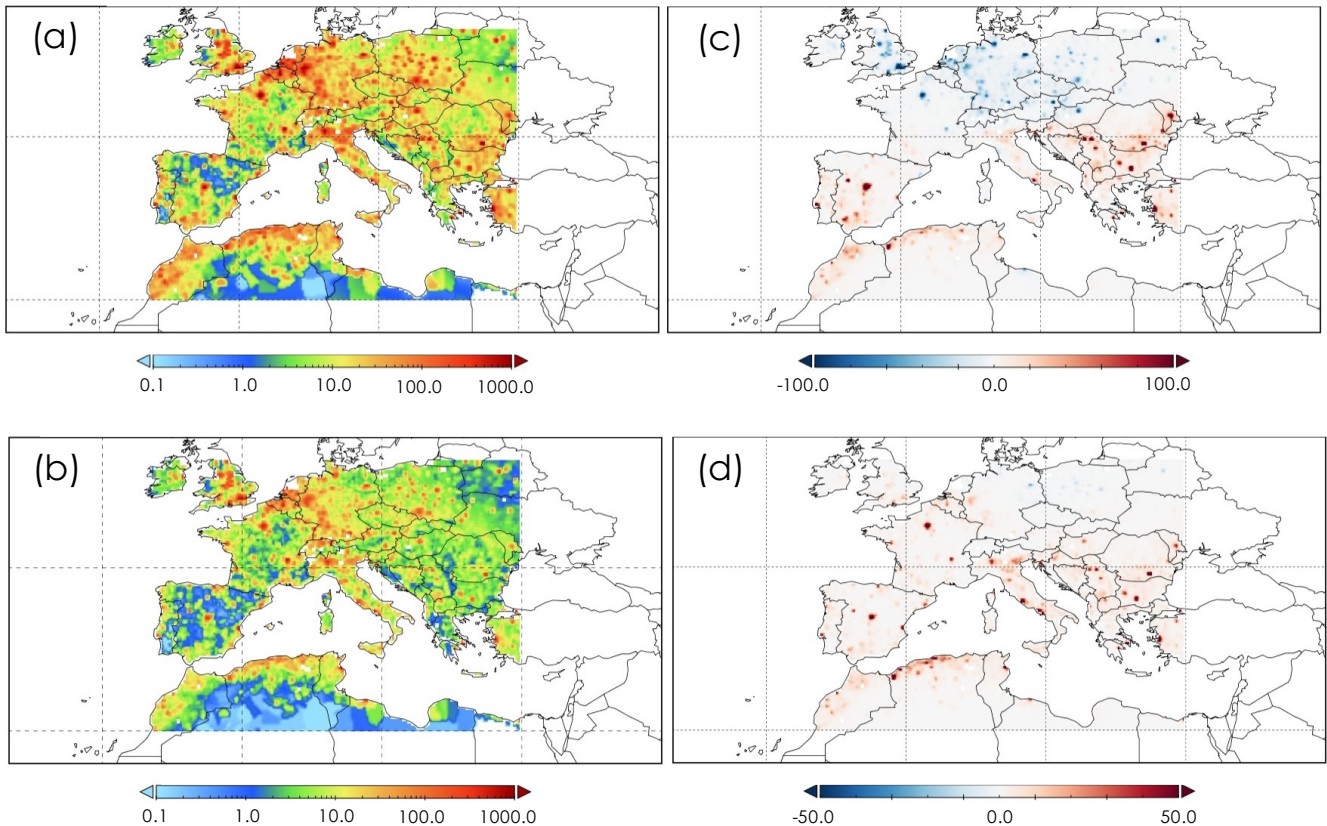

**Figure 7.** (a) Present cases of Premature Deaths (PD) and (b) associated costs, in M€. (c) Changes projected in PD and (d) changes in costs (M€) under the RCP8.5 scenario (2071-2100).

## 4   Conclusions

As proposed in the objectives of this contribution, a relationship was established between air pollutant levels and the impacts
on several human pathologies. Several reasons lead us to carry this study: (1) the exceedances of the limit values regulated by
the European directives or the World Health Organization for several pollutants over some European areas; (2) the scientific
literature available showing a clear and increasing relationship between these exceedances and their impacts on human health
(e.g. Brandt et al. (2013a, b); Im et al. (2018), whose results furthermore support the conclusiones obtained in this work).



The statistical epidemiological study (corroborated later by the modelling results in this same contribution) identify a clear relationship between pathologies and air pollution by PM, especially in central European countries. The highest coefficient of correlation in the epidemiological study are found for total deaths and particulate matter (TD-PM10) in Germany, Slovenia and Czech Republic, and Hungary and Italy for death caused by respiratory diseases (relationship DRD-PM10). The modelling study supports these conclusions, also highlighting that large cities and conurbations (especially in eastern Europe) are to be taken into account in order to analyse the impacts of air pollution on several pathologies and diseases. The pathologies considered importantly impact the societal costs due to the damage produced on population health; and are heterogeneously distributed over Europe and so are the impacts expected due to climate change. Several countries, such as as Moldova or Bulgaria, which are not impacted by present air pollution in the modelled results, will strongly increase the cases and associated costs due to climate change alone.

European megacities are important hotspots for health-related issues due to their traffic, which importantly contributes to CO and $NO_x$ emissions. Moreover, $SO_2$ (included in the estimation of RHA and PD) has a role on pathologies since it comes mainly from energy production. In that case, the impacts on eastern countries are higher due to the high sulfur-content fuels in which they base their economy (Colette et al., 2012; Pozzer et al., 2012; Geels et al., 2015). Henceforth, most of hospital admissions are located in European megacities and eastern Europe, with poor air quality levels causing respiratory damage.

For the future scenario RCP8.5, all pathologies will increase in southern Europe (especially southeastern Europe) because of the changes projected in PM and $O_3$ (this latter related to PD, which are expected to increase in RCP8.5 in the aforementioned areas). On the other hand, northern Europe will benefit from climate change by reducing the levels of air pollution (mainly, PM2.5), as also pointed out by Tagaris et al. (2010). The variation in premature deaths in southern Europe is mainly caused by the increase of $O_3$ due to natural emissions, as a consequence of climate change alone and the accumulation in the Mediterranean of long-range transport of tropospheric ozone (and also particulate matter) (Jiménez-Guerrero et al., 2013a, b). Both pollutants are related with cardiorespiratory diseases and premature death (Tagaris et al., 2010; Geels et al., 2015). A converse behavior is found for more northern areas, where changes in precipitation may reduce the levels of PM (Jiménez-Guerrero et al., 2013b). For this reason, more northern cities such as Berlin or Vienna will benefit from a better air quality on the future projections and a decrease in cases number and, in consequence, on the associated costs.

Premature deaths are the most important pathology in the study area in terms of costs (158 billion €per year, that will increase by 17% in the future RCP8.5 2071-2100 projection) and cases (418700 cases per year increased by 94900 pear year in the future). This has been already stated by several authors (e.g. Héroux et al. (2015)).

Last, we can conclude that, overall, all the pathologies included in this study will increase in the future period under climate change RCP8.5 scenario if no mitigation policies for anthropogenic regulatory pollutants are implemented in Europe. Moreover, we should bear in mind the aging of European population and the increase of city dwellers, variables that have not been taken into account in this study in order just to isolate the effect of climate change alone in the health of European citizens.





*Data availability.* Epidemiological data can be freely obtained from the European Commission page Eurostat (https://ec.europa.eu/eurostat/statistics-explained/index.php?title=Health). The gridded population data has been obtained from the SocioEconomic Data and Applications Center (SEDAC) of NASA (http://sedac.ciesin.columbia.edu). The modelling air quality data generated are accessible upon contact with the corresponding author (pedro.jimenezguerrero@um.es).

5  *Author contributions.* PT-C and PJ-G designed the analysis and wrote the manuscript with contributions from all co-authors; PT-C and MM-S-V compiled the epidemiological data and conducted the statistical analysis; PJ-G conducted the numerical simulations; UI and JB provided access to the EVA model, and together with PT-C and PJ-G estimated cases and costs.

*Competing interests.* The authors declare that they have no conflict of interest.

*Acknowledgements.* The authors acknowledge Project REPAIR- CGL2014-59677-R and ACEX-CGL2017-87921-R of the Spanish Ministry
10  of the Economy and Competitiveness and the FEDER European program for support to conduct this research. The authors also thank the support from the Air Quality Modelling Evaluation International Initiative (AQMEII).



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
