# Peer review of "Isolating the climate change impacts on air pollution-related-pathologies over central and southern Europe - A modelling approach on cases and costs"

_Atmospheric Chemistry and Physics, 2019_

## Referee Comment (RC1) · Anonymous Referee #1 · 15 Mar 2019

This work described the climate change impacts on air pollution-related-pathologies over Europe with a modelling approach on cases and costs. Although this study provides important results and is well written, there remain some concerns in the current manuscript. It would be important to restructure the paper in results/discussion/conclusions, since in conclusions some aspects were discussed, and instead where it should have been discussed in discussion I cannot find almost anything. I strongly recommend the authors to include a limitation section about the applied methods for the epidemiological relationship and modelling of the climate change

scenarios. For instance, at this moment in many regions traffic restrictions are being strained to reduce the pollution exposure.

Major comments:

Pag 2. Line10: I'm missing an important reason. The main issue with pollution is the spatio-temporal variability from local to global scales.

Pag 3. Line 10: "Héroux et al. (2015) suggest that mortality risk associated to air pollution can be reversible on a short period."

In this context, it would be important to difference short and long-term effects. In addition, it cannot be ruled out that reducing the potential exposure also could reduce the risk regarding long-term effects, i. e., the human body can partially recover.

It is also important to highlight that the cost for the health system of the impacts with lower severity and greater population affected can overcome of those situations that have a greater seriousness but a smaller affected population (EEA 2013. Environment and human health, Joint EEA-JRC report Nr 5 Report EUR 25933 EN).

Another aspect would be the vulnerable groups (elderly, people with chronic diseases and children). In an aging society, even if the exposure were reducing, more people would be at risk and vulnerable in the future.

Pag 3. Line 16: Why you use only scenario (2071-2100, RCP8.5)? See Fig. 4 from https://www.atmos-chem-phys.net/18/15471/2018/acp-18-15471-2018.pdf

It could be interesting and useful to include a similar Figure as 5 from the same paper above. The projected changes by different regions in mortality and pollution.

Pag 3. Line 31: Which method you used for detrending? Usually in time series regression, you have to control cofounder variables as temperature, which has significant effects on mortality simultaneously.

See Bhaskaran et al. (2013). Time series regression studies in environmental epidemiology. 10.1093/ije/dyt092 and Analitis et al. (2018). Synergistic Effects of Ambient Temperature and Air Pollution on Health in Europe: Results from the PHASE Project. https://www.mdpi.com/1660-4601/15/9/1856

Pag 4. Line 10: Which is your study area? If Europe, than why you only use most of southern and central Europe?

Pag 5-6. 3.1: Can you discuss with more detail the absence of correlation in some countries, which could be also due to the high spatio-temporal variability and methods issue (in comparison with local epidemiological studies). Discuss limitation of methods in another point and compare you results with city-specific studies.

Pag 15. Line 30: "we should bear in mind the aging of European population and the increase of city dwellers, variables that have not been taken into account in this study in order just to isolate the effect of climate change alone in the health of European citizens."

This is an important aspect and you should discuss it with more detail, in particular, which consequences has this for your results and the limitation of not including a population projection.

Minor comments: Maps. I would suggest using a different projection for Europe. For example, "+proj=laea +lat_0=52 +lon_0=10 +x_0=4321000 +y_0=3210000 +ellps=GRS80 +units=m +no_defs"
* * *

---

## Referee Comment (RC2) · Anonymous Referee #2 · 17 Mar 2019

Comments: • It is not clear to the reader how the authors estimated the present and future costs of air pollution. The authors should explain in detail the methods used to estimate those costs and how they extrapolated their estimates for the future scenario for each country. • The authors used for the epidemiology study a dataset of 25 European countries. Is this compatible with the domain analysed with WRF-Chem? What are the assumptions made here? • One basic limitation here is that the study assumes the population constant for the future scenario. As a robustness check it would help to run a second simulation that takes into account the projected population.

[Figure]

Another limitation of the study is that economy also changes thus affecting air pollution levels. This could also be included in the limitations of the study

---

## Author Comment (AC1) · 3 Jun 2019

A: First, we would like to thank the anonymous referees for their valuable comments in the interactive comment on "Isolating the climate change impacts on air pollution-related-pathologies over Europe – A modelling approach on cases and costs" by Patricia Tarín-Carrasco et al. The manuscript has been revised after the reviewer's comments in order to correct errors and to introduce the reviewers' suggestions for improving the quality of the paper. Please see below our point-by-point replies:

**Anonymous Referee #1**

R#1: This work described the climate change impacts on air pollution-related-pathologies over Europe with a modelling approach on cases and costs. Although this study provides important results and is well written, there remain some concerns in the current manuscript. It would be important to restructure the paper in results/discussion/conclusions […].I strongly recommend the authors to include a limitation section about the applied methods for the epidemiological relationship and modelling of the climate change scenarios […]

A: We strongly appreciate the valuable comments of Reviewer #1. An effort has been made to rewrite the paper, separating the results and discussion section from the conclusions. Moreover, in the conclusions, a brief section has been included trying to highlight the limitations of this work.

R#1: Pag 2. Line 10: I'm missing an important reason. The main issue with pollution is the spatio-temporal variability from local to global scales.

A: We agree with the reviewer's suggestion. This has been added in the revised version of the manuscript.

R#1: Pag 3. Line 10: "Héroux et al. (2015) suggest that mortality risk associated to air pollution can be reversible on a short period."

In this context, it would be important to difference short and long-term effects. […] It is also important to highlight that the cost for the health system of the impacts with lower severity and greater population affected can overcome of those situations that have a greater seriousness but a smaller affected population (EEA 2013. Environment and human health, Joint EEA-JRC report Nr 5 Report EUR 25933 EN). Another aspect would be the vulnerable groups (elderly, people with chronic diseases and children). In an aging society, even if the exposure were reducing, more people would be at risk and vulnerable in the future.

A: We strongly appreciate these valuable comments of Reviewer #1. This has been added in the revised version of the manuscript.

R#1: Pag 3. Line 16: Why you use only scenario (2071-2100, RCP8.5)? See Fig. 4 from https://www.atmos-chem-phys.net/18/15471/2018/acp-18-15471-2018.pdf

A: We strongly appreciate the valuable comments of Reviewer #1. In this work, we used chemistry-climate simulation which covers a part of Europe during 30 of years. This type of simulations, which takes into account atmospheric chemistry and its interactions with meteorological variables, are computationally more expensive than an ordinary climate simulation. Therefore, the RCP8.5 scenario was chosen because it lies at the top of the representative concentration pathways and is estimated to give the largest changes in radiative forcing and then a stronger climatic signal.

R#1: Pag 3. Line 31: Which method you used for detrending? Usually in time series regression, you have to control cofounder variables as temperature, which has significant effects on mortality simultaneously. See Bhaskaran et al. (2013). and Analitis et al. (2018).

A: We strongly appreciate the valuable comments of Reviewer #1. The suggested references, together with further literature revision, have been included in the revised version of the manuscript.

We use the first-differences method for detrending the time series. This has been stated in the manuscript:

*"According to Bashkaran et al. (2013), time series regression studies have been widely used in environmental epidemiology, notably in investigating the associations between exposures such as air pollution and health outcomes. Typically, for both exposure and outcome, data are available at regular time intervals. In our case, an epidemiological study for present situation has been carried out, with data obtained from the European Commission (Eurostat) (https://ec.europa.eu/eurostat/statistics-explained/index.php?title=Health) corresponding to the years 2001-2012. Total Death (TD) and Death caused by Respiratory Diseases (DRD) have been analysed. The objective is to search the correlation between such mortalities and air pollution (in our study case, PM10, due to the short time series available for PM2.5). Although mortality data was available since 1994, the targeted period begins in 2001 due to the availability of PM10 data. As on Analitis et al. (2018), a first order correlation structure was employed"* […] *"The correlation is not done directly on the raw data, but the anomalies of mortality and PM10 series. These series are detrended in order to avoid spurious correlations. The detrending method is based on the first-time difference time series and is widely used in climate data analysis (e.g. Lobell and Field, 2007; Zhao et al., 2017). Linear regressions are performed with first differences in TD and DRD as the response variable, and first differences of PM10 as the predictor variable. The regressions found have undergone a Mann-Kendall test in order to assure their significance at 95% confidence (p<0.05)."*
*"*

However, we are aware about the cofounder variables, but the objective of this study is not to do an epidemiological study. The objective to use these series was to show the presence of a correlation between the pollutans and the pathologies and then to use this correlation to estimate the number of mortality due these pollutants.

R#1: Pag 4. Line 10: Which is your study area? If Europe, than why you only use most of southern and central Europe?

A: We agree with the reviewer's suggestion. The title could lead to confusion because the domain does not cover Europe entirely. For this reason, the title has been modified by : "Isolating the climate change impacts on air pollution-related-pathologies over central and southern Europe – A modelling approach on cases and costs"

R#1: Pag 5-6. 3.1: Can you discuss with more detail the absence of correlation in some countries, which could be also due to the high spatio-temporal variability and methods issue (in comparison with local epidemiological studies). Discuss limitation of methods in another point and compare you results with city-specific studies.

A: We strongly appreciate the valuable comments of Reviewer #1. This has been added in the revised version of the manuscript.

R#1: Pag 15. Line 30: "we should bear in mind the aging of European population and the increase of city dwellers, variables that have not been taken into account in this study in order just to isolate the effect of climate change alone in the health of European citizens."This is an important aspect and you should discuss it with more detail, in particular, which consequences has this for your results and the limitation of not including a population projection.

A: We agree with the reviewer's suggestion. However, the objective of this work is to isolate the effect of climate change to the dwellers wellness. For that reason, we have added the reviewer's suggestion in the revised version of the manuscript.

R#1: Minor comments: Maps. I would suggest using a different projection for Europe. For example, "+proj=laea +lat_0=52 +lon_0=10 +x_0=4321000 +y_0=3210000+ellps=GRS80 +units=m +no_defs"

A: Maps have been re-done by using the suggested projection and included in the the revised version of the manuscript.

**Anonymous Referee #2**

R#2: It is not clear to the reader how the authors estimated the present and future costs of air pollution. The authors should explain in detail the methods used to estimate those costs and how they extrapolated their estimates for the future scenario for each country.

A: We strongly appreciate the valuable comments of Reviewer #2. We followed the methodology of Brand et al. 2013a and 2013b. This is summarized in the manuscript and the formula are compiled on Table 1.

R#2: The authors used for the epidemiology study a dataset of 25 European countries. Is this compatible with the domain analysed with WRF-Chem? What are the assumptions made here?

A: Please see the answer in Reviewer #1's comments. Some of the countries used in the epidemiology study lie out of our simulation domain; that is why the methodology is complementary. In order to avoid confusion, we have changed the name of the manuscript.

R#2: A basic limitation here is that the study assumes the population constant for the future scenario. As a robustness check it would help to run a second simulation that takes into account the projected population. Another limitation of the study is that economy also changes thus affecting air pollution levels. This could also be included in the limitations of the study

A: We strongly appreciate the valuable comments of Reviewer #2. However, this suggestion is slightly beyond the scope of this contribution. The objective of this work was to show how the climate change is going to affect dwellers wellness, hence, we isolate the action of climate change. If population had been changed, it would be difficult to assess the impact of climate penalty itself. That is why we decided to keep population constant.

However, there is another manuscript under writing nowadays (that will complement the present submission) including both the changes in population and the scenario emissions (which take into account changes in the economy).